# A Self-Organizing Multi-Layer Agent Computing System for Behavioral Clustering Recognition

**DOI:** 10.3390/s23125435

**Published:** 2023-06-08

**Authors:** Xingyu Qian, Aximu Yuemaier, Wenchi Yang, Xiaogang Chen, Longfei Liang, Shunfen Li, Weibang Dai, Zhitang Song

**Affiliations:** 1Shanghai Institute of Microsystem and Information Technology, Chinese Academy of Sciences, Shanghai 200050, China; 2School of Physical Science and Technology, Shanghaitech University, Shanghai 201210, China; 3NeuHelium Co., Ltd., Shanghai 200050, China

**Keywords:** field programmable gate array (FPGA), hardware implementation, real-time system, action clustering

## Abstract

Video behavior recognition often needs to focus on object motion processes. In this work, a self-organizing computational system oriented toward behavioral clustering recognition is proposed, which achieves the extraction of motion change patterns through binary encoding and completes motion pattern summarization using a similarity comparison algorithm. Furthermore, in the face of unknown behavioral video data, a self-organizing structure with layer-by-layer accuracy progression is used to achieve motion law summarization using a multi-layer agent design approach. Finally, the real-time feasibility is verified in the prototype system using real scenes to provide a new feasible solution for unsupervised behavior recognition and space-time scenes.

## 1. Introduction

Natural video has the power to be able to serve as unsupervised learning for static as well as dynamic vision tasks. The variation of video frame images contains a large amount of information about the scene and behavior [1,2,3,4,5]. Video data consist of frames, and each frame is a collection of dotted data consisting of RGB pixels. Whether for the recognition of target behaviors or the detection of abnormal behaviors, most of the usual practices first classify the moving targets, such as the mainstream CNN + RNN [6], two-stream [7] or 3D CNN [8], which are then combined with specific feature information for action behavior recognition.

For video processing, the extraction of key points in the original video information requires an intensive computational process that will undoubtedly bring about many computational operations. For instance, the X3D approach [9] typically demands computing power ranging from 6.2 to 48.4 GFLOPs, while the DHTOF architecture based on the RBSOR solver [10] achieves a computing load of 511 GOPS, and consumes approximately 9.4 W of power. Furthermore, the complex and huge computational processes are bound to affect the processing speed. For example, FlowNet [11] uses deep learning to implement optical flow and has a performance of 5–10 FPS. DHTOF achieves excellent processing results but only barely reaches 48 FPS. In an end-to-end real-world application, the too-slow single-frame processing speed cannot keep up with the video frame input rate, and it is difficult to meet the real-time processing. More importantly, in the context of the internet of things (IoT), a large amount of data processing tasks have compelled the shift of computation toward the front-end to alleviate the computational burden. However, such an approach is difficult to implement and deploy in front-end devices due to their limited computing resources and real-time demands for data acquisition. To satisfy task processing at the front-end under limited conditions, it is necessary to address the computational cost and real-time requirements brought about by task processing, especially in the field of video image processing.

In fact, video information is changing continuously, and this changing situation is already strongly regular. By disregarding the classification of moving objects or features, and instead focusing directly on the changing relationships between video frames, it is theoretically possible to achieve clustering and recognition of specific motion behaviors. For example, in the case of a black-and-white silhouette performance, even without knowing the specific identity of the performing character, the background story, etc., it is still possible to understand the meaning of the performance action by watching it. This can be achieved by focusing on the behavior action change situation rather than classifying the target first. The motion states of objects can be learned by summarizing the motion laws to specific types of motion. Then, the collected data in the database are compared for clustering.

In addition, for unlabeled video data, it is not possible to use a priori knowledge to know the detail density and object distribution of the video to be processed, due to the computational processing of the original video. This situation requires either selecting the dataset or completing the filtering manually. The algorithmic model needs to be designed in a targeted way to ensure extraction of feature values from unknown videos. For example, the algorithm model for absolute coordinate calculation applied to the traffic information of a certain traffic road or intersection for targeted processing is difficult to apply directly to another intersection or even another viewpoint without modification, which will lead to difficulties in recognition under the original calculation model.

Meanwhile, the data used by front-end devices usually come directly from continuously collected information and do not have very powerful preprocessing capabilities, especially for tasks such as video tagging. The summarization and learning methods of motion laws also need to consider facing the raw data type of unsupervised processing rather than preprocessed datasets.

In this work, for the traffic road scenario, we propose a self-organizing multi-layer agent computing system (SMLACS) implemented using conventional algorithms. The computational effort is reduced by using a binary coding approach to filter the video information density and retain only motion information. Then, the clustering and identification of motion processes are accomplished using a similarity comparison algorithm. Finally, a self-organizing multilayer structure was designed and used to perform hardware feasibility by combining FPGA with network cameras based on the data collected in real scenarios in real-time (https://github.com/qian-git/MAMMALS, accessed on 28 October 2022). This work aims to provide a new and feasible design idea for motion analysis or even spatio-temporal scene applications. As a result, it achieves basic clustering and recognition capabilities regarding functional effects only. Furthermore, the SMLACS focus on motion information is equally applicable to blurred video data compared to the frame calculation approach that needs finer frames as requirements.

## 2. Related Work

**Clustering recognition.** Clustering is a fundamental pillar for unsupervised motion recognition in the field of computer vision. The mainstream clustering algorithms include three categories: K-means [12], spectral, and hierarchical clustering. The K-means algorithm relies on the value of k, which always needs to be specified before conducting any clustering analysis. For example, Wojciech and Pawel [13] proposed a mixed parallelization of the K-mean algorithm, while Liang et al. [14] used it to handle shape recognition tasks. Li and Hong [15] performed image segmentation processing, while Aditya et al. [16] used fuzzy K-means to complete the adaptive clustering process. Although the K-means algorithm is easy to implement, its disadvantage is that the cluster number K must be predetermined, which is a great challenge for situations where the classification is unknown. Different from the K-means algorithm, spectral clustering is named after the construction of a spectral matrix from a similarity matrix. Spectral clustering converts clustering problems into graph partitioning problems, and identifies the best sub-graphs by searching. For example, Ahmed et al. [17] implemented efficient spectral clustering using Dijkstra’s algorithm, while Wang et al. [18] proposed using message passing and density sensitive similarities to improve spectral clustering. Spectral clustering has a stronger adaptability to data distribution, but its clustering process requires a complete statistical count of individual numbers to effectively establish the spectral matrix. The idea of hierarchical clustering is based on establishing a simple similarity tree between sample pairs. Hierarchical clustering is generally divided into two categories: top-down agglomerative clustering and bottom-up divisive clustering. For example, Xing et al. [19] combined GNN and hierarchical clustering, Saquib et al. [20] improved an efficient parameter-free hierarchy of clusters, and Lin et al. [21,22] used deep CNN in designing hierarchical clustering. The multi-layer agents clustering method proposed in this paper looks similar to top-down hierarchical clustering, but there are essential differences. Hierarchical clustering establishes a clustering tree among a large number of targets, while this paper uses a multi-layer structure to dynamically segment frame images and automatically allocates computing resources to areas with more motion density for fine-grained clustering. Clustering within a single agent is somewhat similar to spectral clustering, where the process matches the best sub-classification.

**Optical flow processing.** In this work, attention is paid to the motion process rather than the specific attributes of objects or background interpretation. In addition to optical flow [23], we are also inspired by MP3 [24], which provides a method for evaluating fixed grid occupancy in continuous frame videos. However, unlike MP3, SMLACS does not calculate all of the dynamic occupancy in the entire frame, which undoubtedly requires huge computational cost, such as the popular optical flow method, and has high computational complexity. Compared to mainstream algorithmic models, SMLACS uses binary encoding to focus only on the motion process, and algorithmic functions are achieved through similarity comparison and minimal logical judgment. By reducing the computational complexity of individual agents via a hierarchical clustering approach with different levels of accuracy, SMLACS can realize clustering of motion behavior at a low computational cost. Such an approach has an additional benefit. For example, the clustering recognition process of SMLACS does not require attention to scene interpretation or object recognition, even scene-based clustering criteria. Furthermore, it is not selective towards video quality and can even handle highly blurred video data.

Most hardware implementations of optical flow have focused on the HS algorithm and the LK algorithm [25]. For example, Lazcano et al. [26] processed 320×240 pixel video frames at 4 FPS by modifying the classical HSOF method on a GPU GeForce NVIDIA-GTX-980-Ti platform. Seong et al. [27] proposed an improved strategy for storing input images after a Gaussian filtering operation using the LK algorithm, and achieved processing 800×600 pixel HD video images at 30 FPS. Li et al. [28] proposed the FPGA-based RAFT-Lite to process video images of 512×396 pixels at a rate of 10.4 FPS, with power consumption reaching up to 13.103 W. Although optical flow-based video processing can achieve high-precision functionality, the associated power consumption and resource usage are difficult to optimize, and real-time processing issues are unavoidable, thus limiting its potential applications. In this paper, the LCS algorithm is used to process optical flow problems with minimal computational operations by transforming them into memory access problems, thereby reducing computational density and workload. The use of DTS encoding further reduces storage capacity requirements, enabling clustering and recognition functions to be achieved with extremely low memory usage. The computational complexity of the system problem is actually managed through a multilayer design that distributes computations across a large number of computational agents, resulting in lower computational complexity for each agent.

## 3. Method

### 3.1. Binary Encoding (BE) and Discrete Temporal Sequence

In terms of focusing on object motion, MP3 [24] proposes an occupancy flow parameterized by the occupancy of the dynamic objects and describes the future occupancy. Inspired by MP3 and in contrast to the optical flow method, which computes precise values to preserve the object feature information, we propose to care only about whether changes occur during the sampling phase of the video information as a basis for encoding and ignore computing specific values. Therefore, based on the background modeling method, we choose to encode binary values (0 and 1) by comparing the difference values. The process of binary encoding using optical flow is shown in Figure 1, where the example and subsequent legends are derived from the webcam capture video frame (https://github.com/qian-git/MAMMALS, accessed on 28 October 2022). The video frame image divides its area into a 4 × 4 data grid, using the current video frame and the background frame to make the difference, and different grids count the different information of the corresponding area. Two parameters are set as the evaluation criteria: the difference size threshold and the number threshold for exceeding the standard difference value, and when the difference data exceed the difference criteria, their statistic is increased by 1. The other cases indicate no change (white areas in the figure).

In this way, the four video frames of the example in Figure 1 are encoded into the corresponding four binary arrays. The different binary arrays under this time slice can capture the motion of an object over time, while there is a logical sequence of changes in the binary data of each grid on the time scale. If there is motion behavior, then the motion pattern can be understood by the change process of the video frame through the binary encoding method. The overall pixel profile of a video frame is not always stable. For example, the overall pixel size may change over time due to ambient lighting. If data are sampled and encoded for a short period of time, a fixed selection of background frames can always be maintained, while for complex environmental conditions or long runs, background frames need to be updated at regular intervals or when the background changes to avoid data failure due to large gaps with the current frame image. Binary encoding enables the conversion of high-density image information into low-density binary array parameters that capture the perception of change. The accuracy of the information data is determined by the number of divided grids, and a denser number of divided grids is set if high-accuracy coding is required.

The binary encoding method enables the transformation of the perceived changes in each grid into a simple encoded form of 0 and 1. Each individual grid can be combined on a time scale into a time series of varying lengths that can represent the duration of change. Only because of the binary encoding, each grid exhibits a continuous process of change that can only be represented as a continuous 1. Figure 2 shows a 4 × 4 uniform grid division and a partial video frame of an encoded slice, after binary, encoding of a car driving. The four grids on the upper right of the image are named A, B, C, and D. The encoded data of the different frames of these four grids are collected to form discrete temporal sequences, and four sets of binary discrete sequences with successive changes are obtained. From the time scale, it is easy to see that there is an obvious correlation between the data changes of the four grids, which is closely related to the vehicle movement process, so it will show the sequential changes shown in the figure.

If only binary encoding is used for the combination of the discrete temporal sequence, not only are more agents required, but also the binary encoding can show a very limited variation and can only indicate the presence or absence of object motion. Moreover, if we only look at the data from a single grid, all of the regions where changes exist will only turn out to be no longer stationary when the picture starts to appear at a certain time, and then return to a stationary state again after a period, and the overall object motion process cannot be summarized by a series of individual grid data. Therefore, the exact design of four adjacent grids of data can use hexadecimal data to superimpose a combination of its encoded values for each frame, which not only improves the data richness that is difficult to express in a binary sequence, but also reduces the information density by a factor of four. The information of consecutive frames compressed in a hexadecimal format still retains the regularity of continuous variation. To further improve the information density, the DTS encoding shown in Figure 3 can be applied, which integrates identical data elements and preserves only the changing ones. Moreover, a character “0” is artificially set as an ending symbol to indicate a complete DTS segment. For example, the fourteen frames of the grid in Figure 2 are combined into “44557FFEEAAA20”, which can be reduced to “457FEA20” again if we ignore the repeated encoding segments and keep only the changed parts. This greatly reduces the information density without losing the effective data.

### 3.2. Similarity Comparison

Binary coding represents the motion process of objects as a discrete temporal sequence. The discrete temporal sequence of this type is stored as a model, and the clustering and identification of the motion behavior of each region can be achieved by comparing the new motion sequence with the similarity of the model. We designed the generic longest common subsequence (LCS) algorithm as the similarity calculation algorithm to obtain the optimal match. The specific calculation process of similarity comparison using the model and the input sequence is performed using the longest common subsequence algorithm approach. The computational process is a bit-by-bit comparison to determine whether the two sequences match. The structure of the LCS algorithm is shown in Figure 4, where the length of the sequence is *i*. In fact, the last term is not necessarily the end marker 0. Here, we just take the complete sequence model as an example, the length of the model is *j*, and the result of the bitwise comparison is counted by iterating through the loop twice. The data between different comparison positions become an inheritance relationship, and finally, the dynamic planning table of (i+1)×(j+1) two-dimensional array data is obtained.

For example, bit *a* of the input sequence of length *i* and bit *b* of the model of length *j* are compared. If they do not agree, the current (a,b) data take the maximum of (a−1,b) and (a,b−1) values; and when their values are the same, the result of the (a,b) position is the value of its diagonal position (a−1,b−1) + 1. The dynamic planning table of the comparison process is statistically obtained in this way. Subsequently, backtracking from the maximum value is used, i.e., backtracking along the path from decreasing values. If the values of (a−1,b) and (a,b−1) that are adjacent to (a,b) are both smaller than 1, the next path position is (a−1,b−1), but if either (a−1,b) or (a,b−1) has the same value as (a,b), the next path position is the position with the same value. Accordingly, the correspondence table of the LCS in the figure is obtained, where the first row represents the specific characters present in both sequences of the LCS, and the second and third rows correspond to the specific position of each column of the first row, corresponding to its data in the input sequence and in the model. For example, if $l_{s2}$ is 3 and $l_{m2}$ is 4, it means that $L_{2}$ is the 3rd in the input sequence and the 4th in the model.

### 3.3. A Single Agent Processing

The processing procedure of a single agent, as shown in Figure 5, includes modules such as agent state management, history encoder, compare core, model library, history state, and model_op. The agent state module mainly handles the working status, configuration, data exchange, and other operations of the agent, enabling, for example, the dynamic configuration of the processing area and the adjustment of which specific areas to handle in multi-level structures. The history encoder module receives input encoding information and carries out DTS coding operations, as shown in Figure 3. The compare core module uses the LCS algorithm for similarity comparison, traverses the model library, and finds the best-matching model, where the model ID is outputted and saved in the history state module for prioritized fast comparison of the next frame’s data. The model library saves all of the learned models of the agent, such as using one BRAM as a model library for each agent in Section 4. The model_op module fuses the current complete DTS features with the adapted model based on the LCS matching result, ensuring the model is updated to follow the changing situation.

### 3.4. Multi-Layer Self-Organizing Structure

BE of the raw video data enables the perception of whether the relevant grid has changed. If the range of the grids is large, which means that the number of grids is small, less information can be obtained through binary coding. For example, if the whole image is not divided, only one grid is stored, so when the overall representation is 1, it only indicates that there is motion in the current frame, but no more detailed information is available. Assuming a pixel-by-pixel division, the detailed shape of the moving object can be obtained through binary coding alone. However, this approach is computationally intensive and loses the advantage of binary coding, so the amount of information available is determined by the size of the divided grid.

However, a video cannot have all of its frames changed, and there will always be some areas where no moving objects always pass by. Furthermore, if the set grid range is small, there may be quite a lot of grids that are always producing invalid data. We propose to use a progressive subdivision layer structure to capture the motion process of the video. The idea of the scheme is to provide more computational resources to the existence of more dense change regions for more detailed data processing. The specific agent system building structure of the scheme is expressed as a layer-by-layer input of agent resources from coarse precision regular summaries to refined regions. According to this dynamic allocation idea, we design a multi-layer structure (MLS) for coarse-to-fine precision progression, and its structure is shown in Figure 6. The green box in the figure indicates the region monitored by an agent, and the yellow divided dashed line in the green box indicates the corresponding four binary encoded regions that receive and process the hexadecimal encoded temporal sequence of the four binary data in the green box by the computing agent.

When an agent keeps a model with multiple temporal sequences, it indicates that there are multiple variations in the region that the agent is responsible for monitoring. This agent then seeks four agents in an idle state to be responsible for monitoring each of the four dashed regions in the figure with a co-working model such as this agent. For example, in the initial operation of the system, the design is shown in Figure 6a that divides the whole screen into only four grids and uses only one agent as the first layer that can be responsible for the whole screen for motion monitoring. The data received by the agent of the first layer is very coarse and is only for determining whether there is a regular change in the four areas of the division but not to obtain more detailed information. Subsequently, the agent of the first layer looks for four agents as the second layer to refine the four binary encoding regions received by the agents of the first layer, and divide the four smaller regions for binary encoding, respectively. Similarly, the third layer is subdivided within the size of the agent management area of the second layer, and the fourth layer is subdivided within the agent management area of the third layer. The agent’s working state switches and the data reception is achieved by filtering the received data for point-to-point transmission. At the same time, the execution process of the agent does not participate in any data exchange behavior except for receiving the input data of each frame at the beginning, which avoids data dependency and improves the processing speed, which can be easily realized to increase the processing speed to the video frame rate.

### 3.5. Summary

Although binary coding and DTS compression reduce accuracy by ignoring a lot of details in the image frames, they focus more on capturing motion information of continuous frames and encode it as a series of discrete sequences. Compared with clustering algorithms, such as K-means that require traversal and multiple calculations, the LCS algorithm can perform similarity comparisons solely through judgment rather than numerical calculation, resulting in faster processing speeds. Additionally, the multi-level structure effectively distributes the computational complexity of video image processing to numerous agent functions, allowing for relatively simple tasks for each agent, making them easier to control. The top-down self-organizing configuration method dynamically allocates computing resources to regions with more frequent changes when facing unknown unsupervised videos, without the need for targeted design. Of course, this self-organizing method also has certain limitations. It encodes the entire region and cannot distinguish between a single target or multiple target scenes, since any change within the corresponding area leads to DTS encoding based solely on binary information, without distinguishing whether simultaneous changes in different regions have associations.

## 4. Prototype System

To verify the feasibility of SMLACS, we designed a prototype verification system. As shown in Figure 7, it consisted of a rich storage resource platform UltraScale + EG board, MIPI camera, and an LCD screen. The UltraScale + EG board is based on Xilinx Zynq−UltraScale’s all programmable SOC. It has four ARM Cortex A53 processors (PS) and 600 K series−7 programmable logic (PL) cells. The PS and PL are connected internally by the AXI bus. The camera’s sampling rate is up to 60 FPS. Furthermore, the board is equipped with 912 blocks of on-chip block RAM(BRAM), each with a capacity of 36 Kb, part of which is delivered to each agent to save and query the sequence model. The computational data of the sampling module of the video images and the subsequent processing of the model output results are stored in the DRAM.

The software interfaces of the prototype system are shown in Figure 8. In the hardware architecture, the MIPI controller outputs the video frames captured by the camera to the system and performs operations such as frame decoding and background differencing of the video frames. Following the frame difference calculation, the original image and differencing results are quickly transferred to the PS through AXI VMDA. The PS binary encodes the differencing results and then transmits them to the agent network to perform clustering recognition. The application receives and aggregates the clustering model recognition results from the agent network and performs discriminative clustering of all of the results into several different types, as well as image labeling and modification of the original image combined with the clustering results.

The internal processing logic of each agent in the prototype system is shown in Figure 9. The agent network receives binary encoded data that flow through each agent’s connector module. If an agent is working, information is recorded in the connector module of that agent about the working mode, the area where the data are received, the level they are in, etc. If the binary-encoded data flow through, they are recorded in the connector module of that agent. If the flow through the binary encoded data matches, the data are saved; otherwise, they are passed on with assistance. Of course, when the subdivision has been carried out in the screen monitored by that agent, the transmitted data are saved and passed on at the same time to ensure that higher-level agents can receive the data. In this way, on the one hand, the data flow can save the transmission overhead, on the other hand, it can be effectively managed. Each agent is internally applied with a separately used BRAM to ensure the efficiency of model queries and to avoid I/O problems caused by the frequent use of mass storage.

## 5. Experiment and Result

### 5.1. Experiment

Our design uses a four-layer structure and requires a fully uniform partitioning case. The design applies 1 + 4 + 16 + 64 computational agents to participate in the computational process, with a total of 85 agents. If the number of layers increases, the maximum number of computational nodes increases exponentially. We used the street view dataset (https://github.com/qian-git/MAMMALS, accessed on 28 October 2022), which was collected from live intersections via a webcam. To achieve an end-to-end unsupervised processing pipeline, a MIPI camera with a resolution of 1280 × 720 pixels was used to capture the frames of the dataset in real-time. During the data collection phase, we recorded the clustering performance displayed on the screen along with the output results and the model saved of each agent post-experiment. Our FPGA system operated at a frequency of 200 MHz, while the camera had a capture rate of 60 FPS. In total, 500 vehicle video samples were tested.

### 5.2. Clustering Performance

Figure 10 shows the effect of the self-organized dynamic segmentation of part of the upper left corner of the screen in Figure 6. (The area is already an agent monitoring range for the second layer). It can be found that as the number of moving vehicles increases, the system gradually and dynamically performs a more fine-grained regular summary based on the changing clustering categories. The agents used for the computation are configured in regions where more variation exists, such as the upper right part of the region displayed in Figure 10, while its upper left part is largely unsegmented, as the region is mostly static. The gray boxes in Figure 10 represent agents configured to handle corresponding areas. For example, Figure 10a illustrates that an agent on the second level divides the area into four grids, and Figure 10b shows that four third-level agents have been configured to perform finer segmentation, and so on. The green markers indicate the successful matching of behavior models for this region in the current frame, accurately identifying them, while red markers indicate recognition errors.

Through the clustering recognition behavior of SMLACS, the output results of the node network after each frame processing can spontaneously build the form of a structure with the multi-layer classification effect. Depending on the clustering results, the classification is performed at different levels of fineness, and the higher the fineness, the more categories there are.

To evaluate the clustering recognition accuracy of the prototype system, a standard for manual classification was employed to categorize the verification dataset into multiple categories. The dataset was classified into three classes based on road and driving style, and further divided into seven classes based on vehicle size. By comparing the saved models and outputs of all agents with the manually classified data, the accuracy of the prototype system was confirmed. In addition, real-time recognition accuracy was evaluated to assess the system’s processing performance. Real-time recognition accuracy refers to the proportion of working agents that correctly identify behavior under the current vehicle video. Only first-level agents cover the entire screen while agents in other levels only process part of the screen area. Therefore, there may be idle agents when some agents work. For example, when vehicles travel on the left side of the road in Figure 6, the agents on the right do not receive any motion information and are not in the working state. Therefore, recognition effectiveness statistics should not consider the condition of idle agents. The clustering results of the first and second level agents in the prototype system are consistent with manual classifications of driving style, with real-time recognition accuracy of 97.4% and 97.3%, respectively. The third and fourth level agents have more classification types because they handle smaller areas and each agent’s clusters change in different areas. As such, calculating classification accuracy for individual agents is not meaningful since different agents deal with different changes in varied areas. Nevertheless, the third-level recognition accuracy reaches 88.5% and the fourth-level recognition accuracy reaches 88%.

### 5.3. Hardware Evaluation

The resource utilization of the prototype system on the FPGA platform is shown in Table 1. Due to the fact that besides implementing the BE + LCS + MLS algorithms, this project also included a real-time output feature for demonstration purposes. We separately displayed the resource usage of both the version excluding the output and that of the whole system. In terms of FFs and LUTs, the actual number of agents used was less than that at the theoretical level because some of the stationary regions were not assigned agents to work. Moreover, the FPGA platform provided an extremely rich set of resources (FFs totaling 548,160, with the utilization rate of 10.38%, and the system utilization rate of 25.49%; LUTs totaling 274,080, with the utilization rate of 31.97%, and the system utilization rate of 58.86%). So, few optimization operations, such as resource reuse, were performed during the software compilation phase.

In the prototype system, the storage space provided by the BRAM was always limited. The design allowed each agent to save up to 16 models, with each model’s length limited to a maximum of 32 bits. During the system operation, assuming all agents had saved the maximum number of models, and using the LCS algorithm for similarity comparison that required traversing all models, even with the assumption that the 85 agents operated almost non-parallelly, the system could still process a single frame in 0.22 ms. As for the system operation count, a maximum of 256 grid areas were need to be encoded for the binary encoding of video frames, in addition to the processing required by the DTS coding, LCS algorithm, and model maintenance modules of the 85 agents. Excluding the screen output display, the estimated operation count of the system was 1.53 MOPs, with a total parameter size of 132.9 Kb, and the system only required 42.5 Kb of BRAM capacity. It can be seen that the operational requirements of the prototype system were far below the amount of hardware resources provided by the FPGA platform, and also much lower than other similar research results of intelligent video processing applications. When compared with the same resource conditions, while meeting the requirements of clustering recognition application, there were still plenty of resources and time available to implement other functional requirements.

Furthermore, the operating power consumption of SMLACS is shown in Table 2. The total power consumption when the platform was running was about 5.059 W, of which the static power consumption was about 0.762 W, accounting for about 15% of the total power consumption. In terms of dynamic power consumption, counting from the type division, the consumption of clocks was about 0.504 W and accounted for 12%; the logic part was 411 mW, which was about 10%; there was about 144 mW consumption for BRAM access, which only accounted for 3%, while the PS side consumed 2.594 W, which accounted for 60% overall. This is because while the algorithm part was processing the analysis, it was also processing and displaying the effect based on the sampled video and clustering results, and this algorithm part consumed a lot of power. The power consumption in terms of filtering out the video output for BE + LCS + MLS is about 1.162 W, where the consumption of a single agent was only 14 mW, while the access consumption of all agents to BRAM was only 76 mW.

The system was instantiated in Verilog on a Xilinx Zynq UltraScale FPGA and designed to operate at a 200-MHz clock rate. Each individual unit was synthesized, placed, and was routed employing the Vivado Design Suite, and was also verified for coherence with the target clock frequency. Given a cycle count of 43.52 K cycles per each frame, the proposed framework could yield an image processing performance of up to 4595 frames/s. Compared to other methods, such as X3D [9], RAFT-Lite [28], and DHTOF [10], that used optical flow for video image processing, our system reduced the number of operations by 3–5 orders of magnitude in computation and lowered the power consumption by approximately 8.1–11.3 times. Admittedly, this work may fall short of achieving a complexity performance similar to the other systems.

## 6. Discussion

### 6.1. Application Scenario

The experimental process demonstrates a self-organizing multi-layer agent system using binary encoding that is verified through an FPGA-based prototype to achieve low-cost computing demands and real-time processing for unsupervised video cluster learning with reduced resource consumption. The proposed solution can be deployed on front-end devices for local processing at coarse precision levels to meet the primary recognition and low-power ongoing operation requirements. Additionally, the resource utilization and temporal overhead of the prototype system are minimal, providing ample headroom for the simultaneous implementation of other functions.

### 6.2. Limitation and Future Directions

This work was designed with a multi-layer structure to meet the self-organizing ability for changing situations, and it operates in a top-down manner regarding video frame coverage. This allows the system to encode any changes in video frames into a form of motion situation. While top-down encoding functions well for encoding and clustering single-object motion, it fails to differentiate multiple-object motion situations as top-level agents encode them as a new motion situation. For example, two people with the same motion process but following one another, will not be recognized as two unknown moving objects, but rather will be encoded as one DTS. The problem of distinguishing multiple-object situations is not only limited to top-level agents, but also each layer of agents cannot distinguish them well due to the unknown occupancy size of moving objects in the video.

Directly evaluating the entire frame is not feasible for handling multi-object situations, which requires fine-grained data processing techniques such as optical flow. One approach involves precision computation and feature extraction similar to other behavior recognition algorithms, while the other involves transforming absolute coordinate motion into relative motion. Two solutions have been conceived based on our current system. Solution 1 is also inspired by the grid occupancy method of MP3 [24]. It extracts object motion information based on the motion changes of consecutive frames, and performs DTS encoding to extract the motion intent of each moving object. Solution 2, different from a top-down approach, adopts a bottom-up hierarchical structure that captures absolute coordinate motion through finer image segmentation at the bottom layer. The movements of different objects can be separately captured based on their absolute coordinate information, enabling multi-object processing on higher layer proxies.

## 7. Conclusions

Video behavior processing poses a challenge in addressing the computational demand, processing frame rate, and power consumption brought about by high-density computing. This paper proposes a motion detection algorithm named SMLACS based on background modeling and optical flow methods. Using binary coding to reduce computational requirements and directly extract inter-frame motion changes, SMLACS summarizes motion information. We designed a prototype verification system based on the FPGA platform and tested SMLACS through real-time processing. Experimental results demonstrate that SMLACS achieves unsupervised video behavior clustering and recognition capabilities well, while significantly reducing computational and power consumption, providing real-time processing capabilities. 

## Figures and Tables

**Figure 1 sensors-23-05435-f001:**
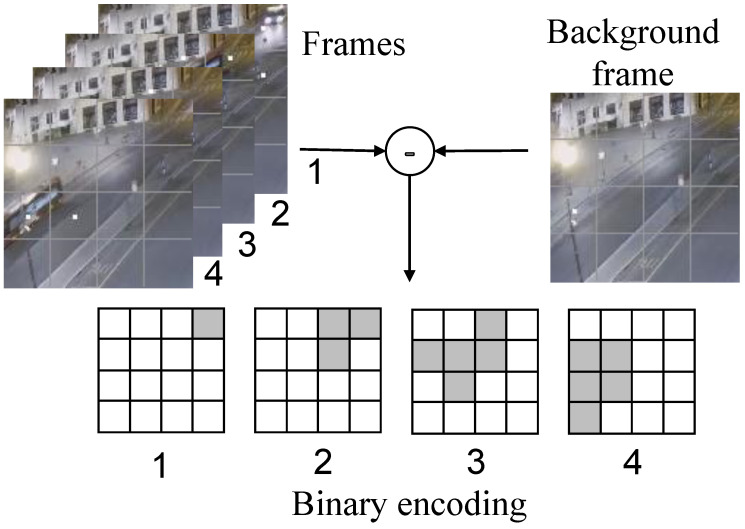
Class optical flow binary encoding.

**Figure 2 sensors-23-05435-f002:**
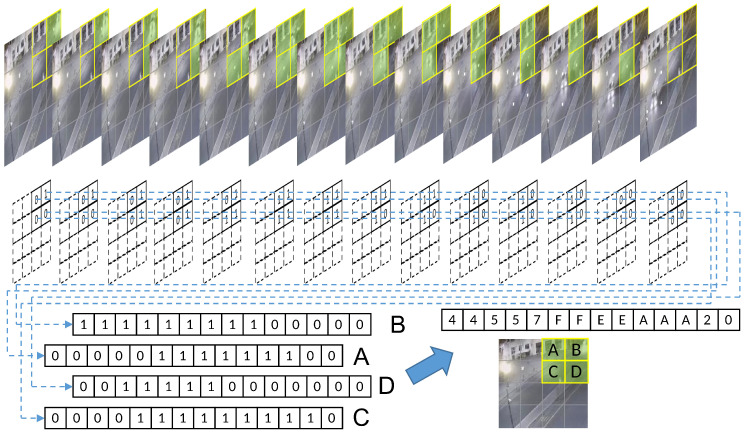
Video frame slicing and discrete temporal sequence coding.

**Figure 3 sensors-23-05435-f003:**
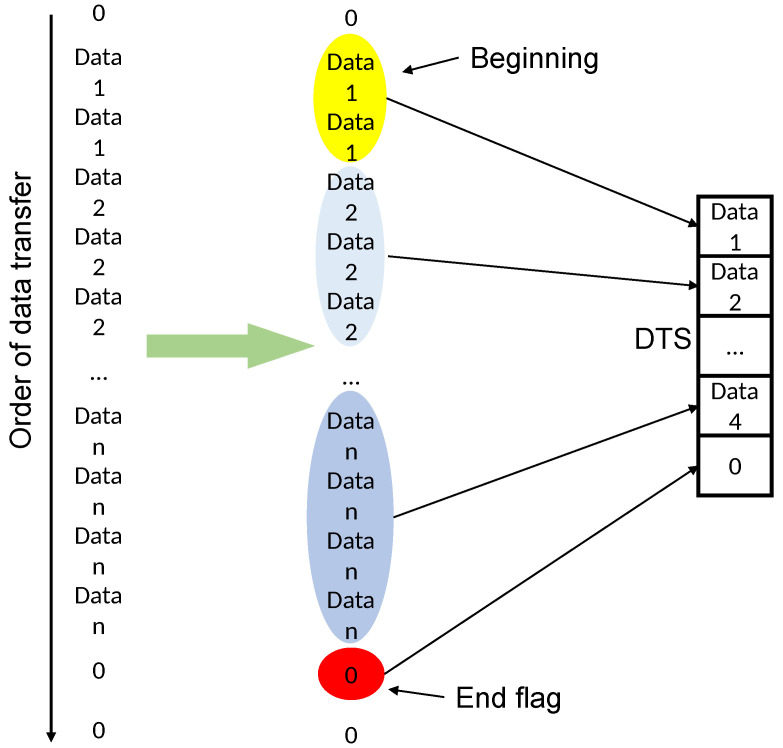
DTS encoding. Data with the same color overlay represents the duration.

**Figure 4 sensors-23-05435-f004:**
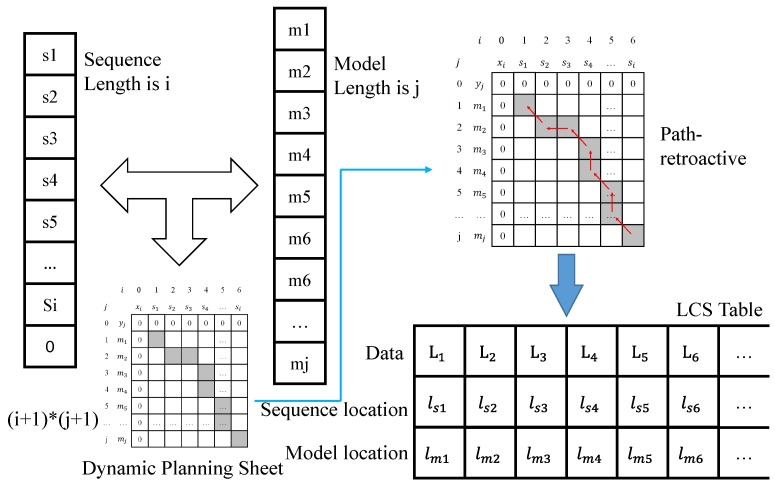
LCS algorithm operation process.

**Figure 5 sensors-23-05435-f005:**
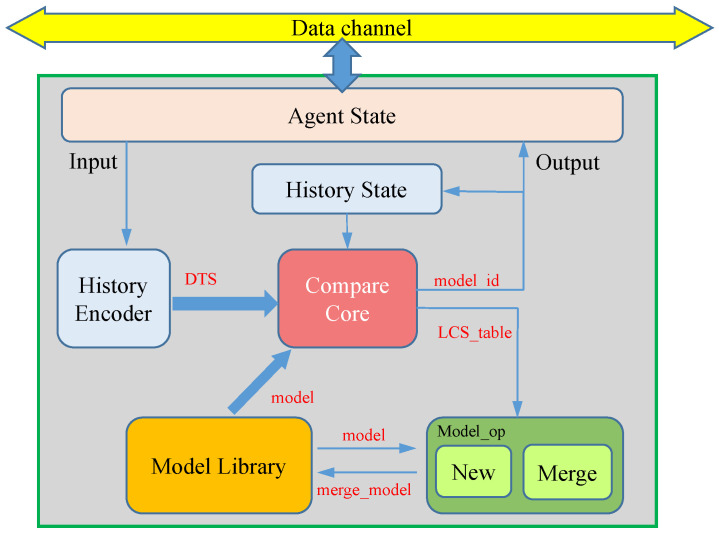
Structure of a single agent.

**Figure 6 sensors-23-05435-f006:**
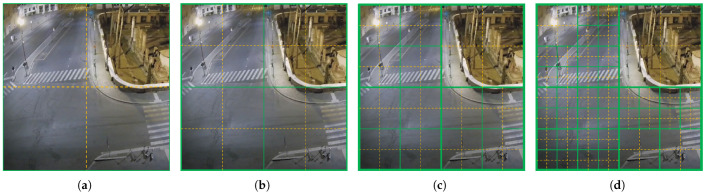
The case of the dynamic segmentation layer by layer. The four-layer dynamic subdivision represented by (**a**–**d**), where (**a**) represents the coarsest accuracy that covers the entire image, and (**b**–**d**) are successively refined subdivisions of the changing area based on the previous layer.

**Figure 7 sensors-23-05435-f007:**
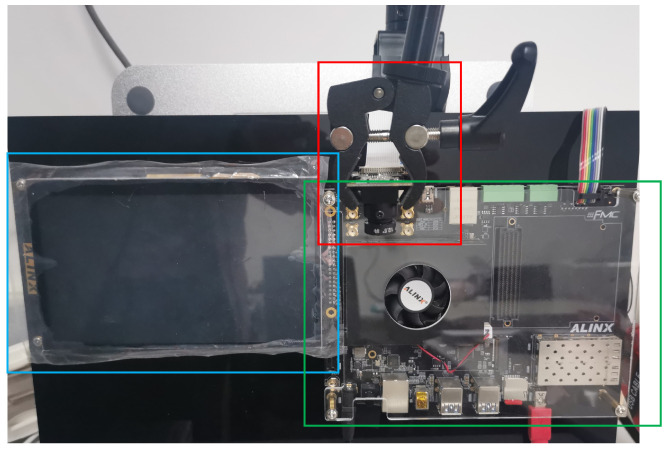
Prototype system. The green box in the picture is the FPGA board, the red box is the MIPI camera, and the blue box is the LCD screen.

**Figure 8 sensors-23-05435-f008:**
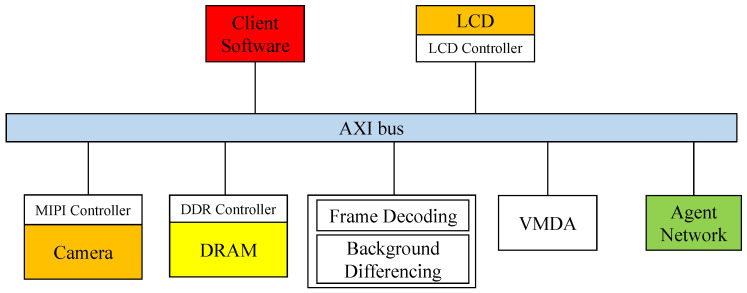
Prototype system working framework.

**Figure 9 sensors-23-05435-f009:**
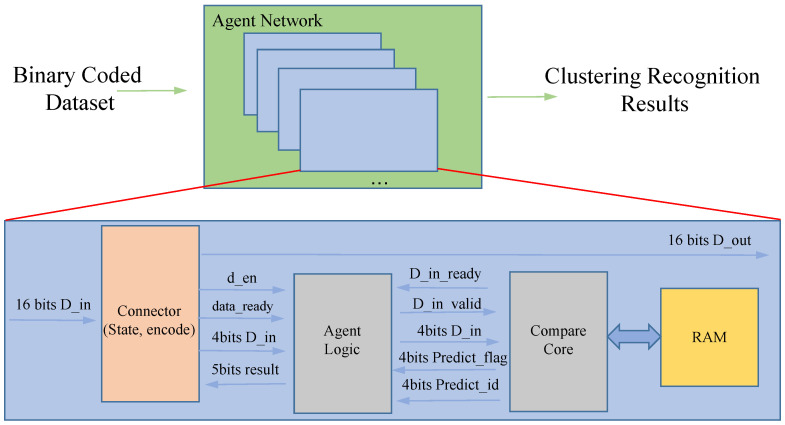
Structure diagram of the agent network operations.

**Figure 10 sensors-23-05435-f010:**
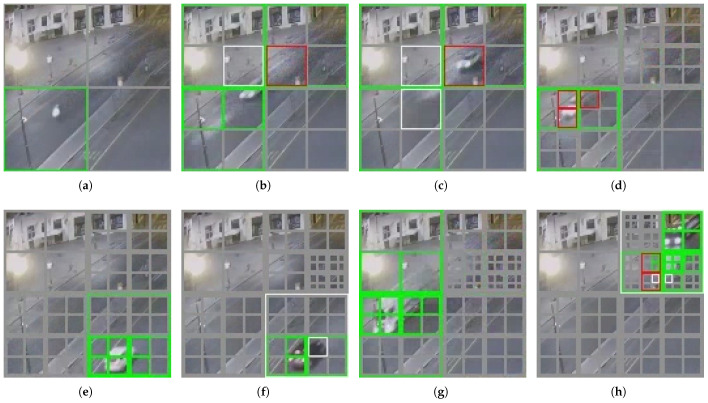
Dynamic division scenarios in operation. (**a**–**h**) represents the real-time dynamic recognition situation in the upper left corner of the screen. Each gray box represents an area covered by an agent, where green indicates correct clustering and red indicates clustering errors.

**Table 1 sensors-23-05435-t001:** FPGA Resource Utilization.

Resource	Available	Utilization	Rate	Utilization	Rate
		(Excluding Output)	(Excluding Output)	(Whole System)	(Whole System)
LUT	274,080	87,612	31.97%	161,327	58.86%
LUTRAM	144,000	−	−	9396	6.53%
FF	548,160	56,910	10.38%	139,699	25.49%
BRAM	912	85	9.32%	161	17.65%
DSP	2520	−	−	8	0.32%
IO	328	68	20.73%	42	12.80%
BUFG	404	2	0.50%	11	2.72%

**Table 2 sensors-23-05435-t002:** Total dynamic and static power dissipation.

		Power	Rate	Power	Rate
Device Static	PS	0.1 W	13%	0.762 W	15%
	PL	0.662 W	87%		
Dynamic Power	Clocks	0.504 W	12%	4.297 W	85%
	Signals	0.425 W	10%		
	Logic	0.411 W	10%		
	BRAM	0.144 W	3%		
	DSP	0.006 W	<1%		
	PS	2.594 W	60%		

## Data Availability

Not applicable.

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
