# Peer review of "A Self-Organizing Multi-Layer Agent Computing System for Behavioral Clustering Recognition"

_sensors, 2023, doi:10.3390/s23125435_

Round 1

Reviewer 1 Report

While the proposed computational system is presented as a new and feasible solution for behavioral clustering recognition in videos, the lack of experimental results limits the analysis of the system's effectiveness. Therefore,  would suggest to add reviewes of similar projects which are running experiments or to present a framework experiment or introduce metrics to evaluate the proposed approach's performance and compare it with other methods to demonstrate the system's advantages.

Additionally, providing more detailed explanations and examples of complex concepts and methodologies would make the document easier to understand for readers who are less familiar with the subject matter. Visual aids, such as figures and diagrams, may also help readers better understand the proposed computational system and how it works.

Also, the document could benefit from a more extensive literature review that provides more context and background information on the topic of video behavior recognition and existing methods and approaches. Expanding the reference list would also help support the proposed approach's novelty and feasibility.

Improvements that should be made to the study to make it more effective and persuasive:

  1. Clearly defining the problem statement and objective of the research to help readers understand the proposed approach's context and motivation.
  2. Providing more detailed explanations of the algorithms and techniques used in the proposed computational system, highlighting their advantages and limitations.
  3. Comparing the proposed computational system's performance metrics, such as accuracy, computational efficiency, and memory usage, with other existing systems or benchmarks.
  4. Discussing potential challenges and limitations of the proposed approach and suggesting possible solutions to overcome them.
  5. Providing a more detailed description of the experimental setup, such as hardware and software requirements, data collection and preprocessing techniques, and evaluation metrics and procedures.
  6. Including additional examples or use cases to help readers understand the practical applications of the proposed approach better.
  7. Concluding the document with a clear summary of the contributions and key findings of the research, highlighting the significance of the proposed approach and the potential impact on the field.

Implementing these improvements would enhance the research study clarity, effectiveness, and persuasiveness, making it a more valuable contribution to the field of video behavior recognition and unsupervised clustering.

The quality of English language in the paper is good overall, with clear and concise sentences, appropriate use of technical terms, and consistent grammar and punctuation. There are a few instances where the sentence structure and wording are not entirely clear, and some sentences could benefit from being rephrased or simplified for clarity. There are also occasional typos or misspelled words, but they do not significantly impact the document's overall coherence or comprehension. Nonetheless, with some minor revisions, the document's language quality could be further improved.

Author Response

Dear reviewer,

We thank the reviewer for reading our paper carefully and giving the above positive comments. These opinions help to improve the academic rigor of our article. Based on your suggestion and request, we have made corrected modifications on the revised manuscript. We hope that our work can be improved again. Furthermore, we would like to show the details as follows:

1. Clearly defining the problem statement and objective of the research to help readers understand the proposed approach's context and motivation.

The authors answer:

Thank for your comments. We have revised the “Introduction” and “Relate work”. In the “Introduction”, additional statements in the paragraph 2 to explain motivation(Line 22-25, Line 26-28, Line 30-37) and additional statements to make the logic clearer (Line 47-49, Line 61-65). In the 'Related Work' section, we have updated and expanded references to cover two main themes, highlighting their novelty. One theme focuses on introducing the differences between our method for cluster recognition and other mainstream algorithms, while the other emphasizes the differences between our optical flow processing solution and alternative methods.

2. Providing more detailed explanations of the algorithms and techniques used in the proposed computational system, highlighting their advantages and limitations.

The authors answer: 

Thank you for your rigorous consideration. We revised the “Method” section to provide more detail and added a “summary” subsection. In the 3.1 subsection, we added the figure 3 and the explanation(Line 221-225), and added a 3.3 subsection to show the operation of the DTS coding and LCS algorithms in an agent(Line262-276). Meanwhile, we added a “summary” subsection(3.5) to highlight the advantages and limitations(Line 320-335).

3. Comparing the proposed computational system's performance metrics, such as accuracy, computational efficiency, and memory usage, with other existing systems or benchmarks.

The authors answer: 

Thank you for pointing out this problem in manuscript. The hardware implementation of mainstream video image processing algorithms is based on Lucas-Kanada and Horn-Schunck. Our research focuses on clustering recognition, making it difficult to conduct direct cross-comparisons. Moreover, our main objective is to explore the reduction of computational complexity and power consumption through our proposed method. Therefore, we modified the experimental section to include two parts: clustering performance (5.2) and hardware evaluation (5.3). In terms of clustering performance, we added an explanation of Figure 10, focusing on the color functionality (Lines 392-396), along with additional performance evaluations of clustering accuracy, including validation and assessment methods, and quantitative data on multilayer clustering accuracy (Lines 403-423), demonstrating good performance in clustering functionality.

As for hardware evaluation, we replaced the old and less relevant references used in Table 1 with hardware resource utilization of the prototype system and modified their corresponding content (Lines 425-436). Additionally, we supplemented performance evaluations of computational operations, processing speed, and memory usage (Lines 444-459), followed by a summary of this section and comparison(Lines 473-482).

4. Discussing potential challenges and limitations of the proposed approach and suggesting possible solutions to overcome them.

The authors answer: 

As Reviewer suggested that we have added a discussion section (Section 6) and provided a detailed explanation of the limitations of the proposed method in the second subsection. Furthermore, we proposed solutions and approaches to address multi-objective problems (Lines 494-515).

5. Providing a more detailed description of the experimental setup, such as hardware and software requirements, data collection and preprocessing techniques, and evaluation metrics and procedures.

The authors answer: 

Thank you for the above suggestions. In the experimental introduction section, we provided additional descriptions of the experiment, including data collection and operation processes (Lines 377-383), and modified and added references to the test dataset (Footnote on Page 1). We also added subsections in the experimental section, namely "Experiment," "Clustering Performance," and "Hardware Evaluation," which improved readability. Performance evaluations were conducted respectively on the statistical methods of clustering accuracy in the "Clustering Performance" section and in the "Hardware Evaluation" section.

6. Including additional examples or use cases to help readers understand the practical applications of the proposed approach better.

The authors answer: 

As Reviewer suggested that in the first subsection of the discussion, we provided an application scope of the study. Specifically, our approach shows promising potential for preliminary identification and screening under low-power conditions (Lines 485-492).

7. Concluding the document with a clear summary of the contributions and key findings of the research, highlighting the significance of the proposed approach and the potential impact on the field.

The authors answer: Considering the Reviewer’s suggestion, we have streamlined the conclusion to enhance its clarity (Lines 517-526).

Extra: The other modifications made in the revised manuscript were for the sake of grammar checking.

Thank you very much for your attention and time. Look forward to hearing from you.

Reviewer 2 Report

The paper describes an activity that appears to be a mixed theoretical engineering and test implementation.

The paper overall is of interest

The theoretical part is somewhat weak: the trick works, but risks and sensitivities for misinterpretation are not sufficiently discussed: it would be helpful to clarify the conditions under which the technique works well, and factors that would risk to compromise its interpretation,and hence the daomian of applicability

For example, it is not clear whether the approach discussed would work for a situation of ships / boats on an agitated sea

the impression is that the technique works well for images with a stable background and limited movement; it may not work on heavy traffic, an overloaded multilane roundabout, nor for maritime applications.

It might have difficulties following a football game . . .

That does not undermine the interest in and applicability of the proposed technique: there are likely a sufficient number of applications meeting the criteria for a succesful usage

the prototype implementation and its results are quite impressive.

the changes that seesm to be necessary are of importance, but it i considered that they should not create a major obstacle for the authors

the recommendation is major revision, to be interpreted as essential but limited in volume 

The English looks good, but semantically could be impoved; an example that may puzzle the reader somewhat:

H&S algorithms require high speed to compute dense and accurate flow vectors with deterministic latency and low power consumption, and thus developed many applications. However, the sequential nature of the iterative solvers in the H&S algorithm leads to long computational processing times for sparse systems of equations until the desired accuracy is obtained.

Author Response

Dear reviewer,

We would like to thank you for your careful reading, helpful comments, and constructive suggestions, which has significantly improved the presentation of our manuscript. Based on your suggestion and request, we have made corrected modifications on the revised manuscript. We hope that our work can be improved again. Furthermore, we would like to show the details as follows:

In the the problem statement and research objectives:

We have revised the “Introduction” and “Relate work”. In the “Introduction”, additional statements in the paragraph 2 to explain motivation(Line 22-25, Line 26-28, Line 30-37) and additional statements to make the logic clearer (Line 47-49, Line 61-65). In the 'Related Work' section, we have updated and expanded references to cover two main themes, highlighting their novelty. One theme focuses on introducing the differences between our method for cluster recognition and other mainstream algorithms, while the other emphasizes the differences between our optical flow processing solution and alternative methods.

In the theoretical Method:

We revised the “Method” section to provide more detail and added a “summary” subsection. In the 3.1 subsection, we added the figure 3 and the explanation(Line 221-225), and added a 3.3 subsection to show the operation of the DTS coding and LCS algorithms in an agent(Line262-276). Meanwhile, we added a “summary” subsection(3.5) to highlight the advantages and limitations(Line 320-335).

In the experimental introduction section:

We provided additional descriptions of the experiment, including data collection and operation processes (Lines 377-383), and modified and added references to the test dataset (Footnote on Page 1). We also added subsections in the experimental section, namely "Experiment," "Clustering Performance," and "Hardware Evaluation," which improved readability.

And we modified the experimental section to include two parts: clustering performance (5.2) and hardware evaluation (5.3). In terms of clustering performance, we added an explanation of Figure 10, focusing on the color functionality (Lines 392-396), along with additional performance evaluations of clustering accuracy, including validation and assessment methods, and quantitative data on multilayer clustering accuracy (Lines 403-423), demonstrating good performance in clustering functionality.

The hardware implementation of mainstream video image processing algorithms is based on Lucas-Kanada and Horn-Schunck. Our research focuses on clustering recognition, making it difficult to conduct direct cross-comparisons. Moreover, our main objective is to explore the reduction of computational complexity and power consumption through our proposed method.  In terms of clustering performance, we added an explanation of Figure 10, focusing on the color functionality (Lines 392-396), along with additional performance evaluations of clustering accuracy, including validation and assessment methods, and quantitative data on multilayer clustering accuracy (Lines 403-423), demonstrating good performance in clustering functionality.

As for hardware evaluation, we replaced the old and less relevant references used in Table 1 with hardware resource utilization of the prototype system and modified their corresponding content (Lines 425-436). Additionally, we supplemented performance evaluations of computational operations, processing speed, and memory usage (Lines 444-459), followed by a summary of this section and comparison(Lines 473-482).

In the discussion and conclusion section:

We have added a discussion section (Section 6). In the first subsection of the discussion, we provided an application scope of the study. Specifically, our approach shows promising potential for preliminary identification and screening under low-power conditions. In the second subsection, we provided a detailed explanation of the limitations of the proposed method, and the approaches to address multi-objective problem.

Also, we have streamlined the conclusion to enhance its clarity (Lines 517-526).

Extra: The other modifications made in the revised manuscript were for the sake of grammar checking.

Thank you very much for your attention and time. Look forward to hearing from you.

Reviewer 3 Report

In this paper, the authors purpose a self-organizing multi-layer agent computing system implemented using conventional algorithms for a particular traffic road scenario.

                The paper is organized on 6 chapters, starting with a synthetic introduction about the field followed by the second chapter where is presented another work related with the present. The purposed method is described in chapter 3 presenting the binary encoding in a discrete temporal, similarity calculation algorithm and the structure of multi-layer self-organizing. To verify the method purposed, the authors designed a prototype also is used to obtained the results presented in chapter 5. The paper end with the conclusions.

I have the following observations:

-          Due to the technologies evolution a part of references is older;

-          The state of the art presented in table 1 need to have a common point of comparation because, for example, the frame rate is not relevant because of different technologies used older or newer;

-          The method purposed is presented on a very particular scenario, more than that, in the images is moving only one object, what is happening when the street became crowded at a moment, or two objects interests the trajectory?

-          Please, highlight the novelty with the related works.

Author Response

Dear reviewer,

We would like to thank you for your careful reading, helpful comments, and constructive suggestions, which has significantly improved the presentation of our manuscript. Based on your suggestion and request, we have made corrected modifications on the revised manuscript. We hope that our work can be improved again. Furthermore, we would like to show the details as follows:

1. Due to the technologies evolution a part of references is older;

The authors answer: 

Thank you for the above suggestions. In the revised manuscript, we have updated and expanded the references in recent years to make them more relevant to the manuscript.

2. The state of the art presented in table 1 need to have a common point of comparation because, for example, the frame rate is not relevant because of different technologies used older or newer;

The authors answer: 

Thank you for pointing out this problem in manuscript. The hardware implementation of mainstream video image processing algorithms is based on Lucas-Kanada and Horn-Schunck. Our research focuses on clustering recognition, making it difficult to conduct direct cross-comparisons. Moreover, our main objective is to explore the reduction of computational complexity and power consumption through our proposed method. Therefore, we modified the experimental section to include two parts: clustering performance(5.2) and hardware evaluation(5.3). In terms of clustering performance, we added an explanation of Figure 10, focusing on the color functionality(Lines 392-396), along with additional performance evaluations of clustering accuracy, including validation and assessment methods, and quantitative data on multilayer clustering accuracy(Lines 403-423), demonstrating good performance in clustering functionality.

As for hardware evaluation, we replaced the old and less relevant references used in Table 1 with hardware resource utilization of the prototype system and modified their corresponding content(Lines 425-436). Additionally, we supplemented performance evaluations of computational operations, processing speed, and memory usage(Lines 444-459), followed by a summary of this section and comparison(Lines 473-482).

3. The method purposed is presented on a very particular scenario, more than that, in the images is moving only one object, what is happening when the street became crowded at a moment, or two objects interests the trajectory?

The authors answer: 

We gratefully appreciate for your valuable suggestion. For a detailed explanation of the use and limitations of the method, we added a “summary” subsection(3.5) to highlight the advantages and limitations(Line 320-335). And we have added a discussion section (Section 6). In the first subsection of the discussion, we provided an application scope of the study. Specifically, our approach shows promising potential for preliminary identification and screening under low-power conditions. In the second subsection, we provided a detailed explanation of the limitations of the proposed method, and the approaches to address multi-objective problem.

4. Please, highlight the novelty with the related works.

The authors answer: 

Thank you for pointing out this problem in manuscript. In the 'Related Work' section, we have updated and expanded references to cover two main themes, highlighting their novelty. One theme focuses on introducing the differences between our method for cluster recognition and other mainstream algorithms, while the other emphasizes the differences between our optical flow processing solution and alternative methods.

Other modifications:

Introduction:

In the “Introduction”, additional statements in the paragraph 2 to explain motivation(Line 22-25, Line 26-28, Line 30-37) and additional statements to make the logic clearer (Line 47-49, Line 61-65).

Method:

We revised the “Method” section to provide more detail. In the 3.1 subsection, we added the figure 3 and the explanation(Line 221-225), and added a 3.3 subsection to show the operation of the DTS coding and LCS algorithms in an agent(Line262-276).

Experiment and Result:

We provided additional descriptions of the experiment, including data collection and operation processes (Lines 377-383), and modified and added references to the test dataset (Footnote on Page 1). We also added subsections in the experimental section, namely "Experiment," "Clustering Performance," and "Hardware Evaluation," which improved readability.

And we modified the experimental section to include two parts: clustering performance (5.2) and hardware evaluation (5.3). In terms of clustering performance, we added an explanation of Figure 10, focusing on the color functionality (Lines 392-396), along with additional performance evaluations of clustering accuracy, including validation and assessment methods, and quantitative data on multilayer clustering accuracy (Lines 403-423), demonstrating good performance in clustering functionality.

Conclusion:

We have streamlined the conclusion to enhance its clarity (Lines 517-526).

Extra: The other modifications made in the revised manuscript were for the sake of grammar checking.

Thank you very much for your attention and time. Look forward to hearing from you.

Round 2

Reviewer 1 Report

The paper is improved significantly and includes detailed figures and diagrams to illustrate the structure and operation of the prototype system and experimentation. The inclusion of detailed diagrams, coupled with the thorough exposition of the experimental procedures, significantly enhances the study and also reader's understanding  of the research undertaken.